# The Hormonal Background of Hair Loss in Non-Scarring Alopecias

**DOI:** 10.3390/biomedicines12030513

**Published:** 2024-02-24

**Authors:** Barbara Owecka, Agata Tomaszewska, Krzysztof Dobrzeniecki, Maciej Owecki

**Affiliations:** 1Students’ Scientific Association in Public Health, Poznań University of Medical Sciences (PUMS), Święcickiego 6, 60-781 Poznań, Polandkd34610@gmail.com (K.D.); 2Department of Public Health, Poznań University of Medical Sciences (PUMS), Święcickiego 6, 60-781 Poznań, Poland

**Keywords:** hair loss, androgens, estrogens, thyroid hormones, growth hormone, corticotropin-releasing hormone, androgenetic alopecia, polycystic ovary syndrome, alopecia areata, 5-alpha-reductase

## Abstract

Hair loss is a common clinical condition connected with serious psychological distress and reduced quality of life. Hormones play an essential role in the regulation of the hair growth cycle. This review focuses on the hormonal background of hair loss, including pathophysiology, underlying endocrine disorders, and possible treatment options for alopecia. In particular, the role of androgens, including dihydrotestosterone (DHT), testosterone (T), androstenedione (A4), dehydroepiandrosterone (DHEA), and its sulfate (DHEAS), has been studied in the context of androgenetic alopecia. Androgen excess may cause miniaturization of hair follicles (HFs) in the scalp. Moreover, hair loss may occur in the case of estrogen deficiency, appearing naturally during menopause. Also, thyroid hormones and thyroid dysfunctions are linked with the most common types of alopecia, including telogen effluvium (TE), alopecia areata (AA), and androgenetic alopecia. Particular emphasis is placed on the role of the hypothalamic–pituitary–adrenal axis hormones (corticotropin-releasing hormone, adrenocorticotropic hormone (ACTH), cortisol) in stress-induced alopecia. This article also briefly discusses hormonal therapies, including 5-alpha-reductase inhibitors (finasteride, dutasteride), spironolactone, bicalutamide, estrogens, and others.

## 1. Introduction

Hair loss is a common condition affecting both men and women and impacting the majority of the population by age 80 [1]. Given that head hair plays a crucial role in the perception of an ideal physique, hair loss can cause serious psychological distress [2]. Studies have shown the effects of alopecia on reduction in self-esteem, difficulties in coping with stress, and the increased incidence of depression and anxiety disorders [3,4,5]. Emotional distress related to hair loss may occasionally be extremely intense, with cases of suicide linked to hair loss that have been reported [6]. This risk is particularly significant in patients with concomitant endocrinopathies, as these conditions may themselves cause psychiatric disorders [7].

The hair growth process consists of three main phases: anagen, catagen, and telogen, constituting growth, transition, and rest, respectively [8]. Some authors additionally distinguish an exogen phase, during which hair shaft release is initiated and successfully executed [9]. The hair cycle entails a complex interplay of various endocrine, autocrine, and paracrine signaling pathways [10]. Hormones significantly impact the structure of hair follicles and hair growth by influencing the hair cycle. Most notably, hair condition is affected by the action of androgens, which bind to intracellular androgen receptors in dermal papilla cells to affect the hair follicle. The impact of androgens on hair follicles varies depending on the location of the hair on the body [11]. A genetically predisposed elevated sensitivity of hair follicles to androgens leads to androgenetic alopecia—the most prevalent form of nonscarring hair loss [12]. Also, the impact of thyroid hormones (THs) on hair growth has been a subject of particular study, with previous research providing strong evidence linking THs to alopecia. In addition to androgens and thyroid hormones, the impact of other hormones, such as estrogens, corticotropin-releasing factor (CRF; also known as corticotropin-releasing hormone, CRH), adrenocorticotropic hormone (ACTH), cortisol, and growth hormone (GH), on the process of hair growth, is currently being studied [11]. Various hormonal disorders facilitate the shift from the anagen to the telogen phase, potentially resulting in hair loss [13,14]. Considering the frequency of endocrine disorders and the possibility that hair loss might serve as their sole clinical manifestation, it is crucial to consider hormonal causes in the diagnosis of alopecia thoroughly.

Despite the widespread occurrence of hair loss, its heterogeneous etiology contributes to limited diagnostic and therapeutic options in many cases. Hormonal changes play a key role in the pathogenesis of many types of alopecia [15]. Hence, a review of the current literature for endocrine causes is important to continuously update the comprehensive clinical management of hair loss. This work offers current insight into hormonal impacts on alopecia and highlights some of the emerging therapeutic options related to this condition.

## 2. Androgens

Androgens are one of the most important factors promoting hair growth both in men and women [16,17]. The primary mechanism of their action involves binding to the intracellular androgen receptor (AR) in dermal papilla cells (DPCs), which initiates a signaling cascade, leading to the transformation of thin, short, and straight vellus hair into dark, longer, and curly terminal hair. This process occurs in androgen-dependent regions, including axillary and pubic areas in both sexes and the beard and trunk areas in men [16,17]. Human androgens are produced mainly in gonads and adrenal glands, as well as in other organs, including skin, adipose tissue, and the brain [18]. The most important androgens synthesized in gonads are testosterone (T) and 5α-dihydrotestosterone (DHT), whereas in adrenal glands, 4-androstenedione (A4), dehydroepiandrosterone (DHEA), and its sulfate DHEAS are synthesized [18,19]. T and DHT are the principal and most potent androgens involved in the hair growth process. T is the main androgen circulating in the blood, while DHT is produced during the peripheral conversion of T by the enzyme 5α-reductase. Compared to T, DHT has a greater affinity for AR. Namely, DHT binds to the AR approximately 4 to 5 times more strongly and dissociates 3 times slower than T [12,20,21,22].

Although DHT promotes hair growth in androgen-dependent regions, its action in hairy scalp skin is quite the opposite. Here, evidence shows that one of the causes of hair loss is excess DHT. Increased activity of type II 5α-reductase, and thus an excessive production of DHT, was observed in the hair follicles (HFs) of people with androgenetic alopecia (AGA) [12,22]. AGA can be divided into male androgenetic alopecia (MAGA), otherwise known as male-pattern hair loss (MPHL), and female androgenetic alopecia (FAGA), otherwise known as female-pattern hair loss (FPHL) [23,24]. Excess DHT causes the miniaturization of hair, reducing the anagen phase and increasing the telogen phase, leading to hair loss and decreased hair density [25]. Typically, MPHL affects the frontal scalp and vertex, and its characteristic feature is a receding hairline [26]. Occipital hairs are usually insensitive to androgen excess due to AR methylation in DPCs located in the occipital region [12,22]. The assessment of androgen levels is possible in serum, saliva, and urine samples. However, the clinical interpretation of the measurements is complicated because most androgens are bound to proteins, and, therefore, their action is not directly related to their total serum concentrations. Some assays enable the measurement of free T; however, to avoid laboratory inaccuracies, it is better to measure total testosterone and the concentration of sex hormone-binding globulin (SHBG) [27].

Hair loss may also be one of the signs of androgen excess in women, along with hirsutism and acne. FPHL affects the central parietal part of the scalp [28]. In most cases, it is connected with polycystic ovary syndrome (PCOS) [29]. However, the exact diagnosis may sometimes be challenging, as late-onset non-classic congenital adrenal hyperplasia (NC-CAH), due to 21-hydroxylase deficiency, may mimic the clinical and laboratory signs and symptoms of PCOS. It is worth noting that the key laboratory marker differentiating those two clinical entities is 17-hydroxyprogesterone (17-OHP), which is a precursor of A4. It is usually normal or slightly elevated in PCOS but significantly higher in NC-CAH [30]. Androgenetic alopecia is reported in 42.5% of women with PCOS compared to 6% in the general female population under 50 years [31]. In understanding the pathogenesis of androgen excess in women, one has to consider that the sources of these hormones are different in men and women. In contrast to testes in men, the ovaries, adrenal glands, and peripheral tissues in women are responsible for androgen production and metabolism. Dehydroepiandrosterone (DHEA), its sulfated ester DHEAS, and androstenedione (A4) are produced in the adrenal cortex and converted into T and DHT in peripheral tissues and the ovaries [29]. Typically, in PCOS-related AGA, mild-to-moderate elevations in serum T, A4, and/or DHEAS are observed. However, it is worth mentioning that serum T is a parameter susceptible to various factors and dependent, among others, on the concentration of SHBG. Therefore, T levels are not always elevated in women with androgenetic alopecia. It has been shown that 25% of women presenting androgen excess symptoms have normal T levels but elevated A4 levels. A4, unlike T, is less dependent on SHBG concentration [29,32]. Interestingly, AGA severity in women is negatively correlated with serum SHBG levels, which is explained by the fact that higher SHBG concentrations bind more testosterone, lowering its free and active fraction. Consequently, this decreases the negative impact of T on hair growth [33]. For this reason, total T measurement should be performed only in line with the SHBG test, which helps estimate the amount of free testosterone. Of note is that commercially available free testosterone assays have low sensitivity, and we do not recommend their use. Therefore, if a hormonal cause of hair loss in women is suspected, the following parameters should be determined in the early follicular phase (between days 2 and 5 of the menstrual cycle): LH, follicle-stimulating hormone (FSH), estradiol, T, DHEAS, A4, SHBG, 17-OHP, prolactin and thyroid function tests [29].

Interestingly, some researchers suggest that there is a male equivalent of PCOS, and one of the diagnostic criteria for this syndrome is early-onset AGA. To diagnose early-onset AGA, alopecia must be equal to or greater than III grade in the Hamilton–Norwood scale and occur at under 35 [34]. The Hamilton–Norwood scale is a seven-grade classification of the severity of hair loss. At stage III, the first signs of clinically significant balding appear. According to the original description by Norwood et al., most type III scalps have deep frontotemporal recessions, which are usually symmetrical and are either bare or very sparsely covered by hair. These recessions extend further posteriorly than a point which lies approximately 2 cm anterior to a coronal line drawn between the external auditory meatus [35,36]. Men with early-onset AGA may have increased DHEAS levels, decreased SHBG levels, and worse gonadal steroidogenesis [34]. Increased DHEAS, along with 17-OHP, is a determinant of increased adrenal activity [37,38]. For details, see Table 1.

The US Food and Drug Administration (FDA) approved two drugs for AGA treatment, namely, topical minoxidil and oral finasteride; however, the latter one is only approved for MAGA. The action of finasteride is strictly connected with androgens, as it is a 5α-reductase type II inhibitor that blocks the production of DHT. A daily dose of 1 mg oral finasteride in MAGA enables a reduction in scalp and serum DHT by 60% [39]. It also shows effectiveness in FPHL, especially when used at a higher dosage (5 mg/d) for at least a year. However, despite its efficacy, it is not approved by the FDA in women due to its adverse hormonal effects [40]. Hence, first-line therapy in FPHL includes 5% minoxidil, a non-hormonal vasodilator. In second-line therapy, finasteride and other antiandrogenics may be considered off-label, especially when hair loss is severe or in women with hyperandrogenism [41]. Another 5α-reductase inhibitor, which is used off-label in AGA treatment, is dutasteride. A recent meta-analysis showed that 0.5 mg of dutasteride daily is more effective in AGA treatment than 1 mg of finasteride daily. However, this difference disappeared after a higher dosage of finasteride (5 mg daily) [42]. There are also other non-FDA-approved treatments that show effectiveness in patients with AGA. These include, among others, hormonal therapies with antiandrogenic effects: oral spironolactone, bicalutamide, cyproterone acetate (CA), and topical androgen receptor inhibitors: clascoterone and pyrilutamide [23,26,43,44]. Spironolactone, a synthetic aldosterone receptor antagonist, also acts as an AR antagonist and decreases total testosterone levels by blocking adrenal enzymes involved in the synthesis of T (17α-hydroxylase and desmolase) [45,46]. It is used in the treatment of FPHL with an overall hair loss improvement rate reaching 56.60%, as reported in a recent meta-analysis. Importantly, better results were obtained in a combined therapy group (oral spironolactone and minoxidil) in comparison with monotherapy (65.80% and 43.21%, respectively) [47]. The standard dose of spironolactone is 100–200 mg daily; however, doses range from 25–200 mg daily [26]. Bicalutamide is an emergent selective AR antagonist with favorable safety and reduction in hair loss severity by 24.5% at a 1-year measurement (measurement in Sinclair scale). Studies suggest efficient doses are 10–50 mg daily [48,49,50]. There are also natural antiandrogens proposed in AGA therapy. One of them is pumpkin seed oil, which blocks 5α-reductase. Mean hair count increased by 40% in the group of men treated with pumpkin seed oil at 24 weeks, compared with 10% in the placebo-treated group [21,51,52]. Moreover, topical pumpkin seed oil did not have significant adverse effects in comparison with the side effects of finasteride and minoxidil [53,54]. For details, see Table 2.

## 3. Estrogens

Human estrogens include estrone, estradiol, and estriol, with estradiol being the most potent. Estrogens exert their effects through two receptors, ER-α and ER-β, with the latter being observed in hair follicles [55]. Binding with ER-β, estrogens can substantially modify the growth and life cycle of the hair follicle by exerting influence on the aromatase activity responsible for converting androgens into estrogens. Particularly, estradiol extends the anagen phase of the hair cycle, amplifying hair growth through the augmented synthesis of growth factors that stimulate the proliferation of keratinocytes of the outer root sheath [56]. Consequently, estrogens are assumed to promote hair growth, increase hair diameter, and prevent hair loss [13].

A decrease in estrogen levels below their physiological concentration is known as hypoestrogenism. Various conditions can result in hypoestrogenism, including menopause (the most common cause), premature ovarian failure, and other causes of hypogonadism [57,58,59]. In previous studies, the relationship between menopause and hair loss has been studied and described in particular [60,61]. Estrogens in premenopausal non-pregnant women are mainly synthesized in the ovaries. After menopause, when the hormonal activity of the ovaries is lost, the main site of estrogen production shifts from the ovaries to peripheral tissues such as the skin, adipose tissue, bone, and brain. For that reason, amid all hormonal changes associated with menopause, one central process that occurs is a significant decline in estrogen levels [62]. In line with this, the protective role of estrogens against hair loss has been assumed based on phenomena of diminished hair renewal, growth, thickness, and hair rarefaction observed during the menopausal period [56]. This function of estrogens is further supported by the observation that the frontal hairline typically remains unaffected in female pattern androgenetic alopecia; this is explained by a relatively high level of aromatase in the frontal hairline region of the scalp. Aromatase is the enzyme responsible for converting testosterone to estradiol and androstenedione to estrone; thus, high levels of this enzyme lower the amounts of androgens and enrich the estrogen supplies to this part of the skin [63]. Nevertheless, it should be noted that the decline in estrogen levels associated with menopause induces hair loss only in some women. Other factors affecting hair health include diet, stress, genetic factors, chronic health issues, use of medications, and nutritional deficiencies [64]. Additionally, the estrogens-to-androgen ratio may be responsible for alopecia in female-pattern hair loss rather than the absolute values of either hormone [65]. These findings require further studies.

Considering the above-mentioned physiological background, hormonal therapies containing estrogens are believed to exert independent regulatory effects on hair growth and to address androgen-dependent disorders through indirect mechanisms that impede androgenic actions. These mechanisms involve the elevation of sex hormone-binding globulin levels, consequently diminishing the availability of androgens [66]. Improvement in the frontal hairline thinning score and increase in the plucking strength resulting from 6 months of hormone replacement therapy was revealed [67]. Nevertheless, precaution is imperative in the prescription of hormonal therapies, as estrogens have been linked to heightened susceptibility to cardiovascular events, stroke, venous thromboembolic events, and breast cancer [66].

The management of alopecia may be particularly complex in women diagnosed with breast cancer, being the most common malignancy in women globally [68]. Hormonal therapy currently constitutes the basic treatment of early-stage breast cancer. Tamoxifen (acting as a selective estrogen receptor modulator), as well as aromatase inhibitors and ovarian function suppression (which lower estrogen levels), administered individually and in combination, significantly reduce recurrence and mortality [69]. The inhibition of androgen-to-estrogen conversion may lead to oncotherapy-induced alopecia, which clinically resembles female androgenetic alopecia. Additionally, hair loss can be exacerbated by the high-stress level associated with a cancer diagnosis. Moreover, due to the elevation of sebaceous glands secretion, the texture and appearance of the hair changes. Treatment options for oncotherapy-induced alopecia are limited, as 17β-estradiol should not be used in these patients due to its ability to bind to estrogen receptors. On the other hand, 17α-estradiol does not bind to estrogen receptors but may increase the conversion of testosterone to 17β-estradiol, presumably by increasing aromatase activity; therefore, this treatment may also pose potential risks to patients with hormone receptor-positive breast cancer [70]. Therefore, in the potential treatment of alopecia in breast cancer patients, spironolactone, acting as an antagonist of the androgen receptor (as described above), might be considered. The potential safety of spironolactone may be attributed to the fact that it has been shown not to increase estrogen levels in most patients. Studies involving 49,298 patients did not consistently show evidence of an elevated risk of female breast cancer associated with the use of spironolactone [71].

Hair loss linked to estrogenic activity may also be a manifestation of COVID-19. Estrogens demonstrate anti-inflammatory effects through the inhibition of pro-inflammatory cytokines. During the infection, the depletion of estrogens diminishes their protective function on hair follicles, contributing to hair loss [72]. Further research on the correlation between COVID-19 as well as other diseases affecting the immune system, with a particular focus on postmenopausal women with permanent reduction in levels of estradiol, is needed.

## 4. ACTH, CRH, and Cortisol

Evidence shows that stress can induce hair loss by acting through stress hormones [73,74,75,76,77]. Corticotropin-releasing factor (CRF), adrenocorticotropic hormone (ACTH), and cortisol belong to the main stress hormones in humans and constitute the elements of the hypothalamic–pituitary–adrenal axis [78]. CRF is secreted by the paraventricular nucleus of the hypothalamus under numerous stressors and plays a pivotal role in the regulation of the stress response, as it activates the HPA axis and promotes the synthesis of ACTH by the pituitary gland. ACTH stimulates the adrenal cortex to produce and release glucocorticoids, such as cortisol [78,79]. Receptors for cortisol and CRF are found, among others, in the skin [74]. Interestingly, some researchers suggest that human DPCs have a fully functional HPA axis and are able not only to respond to CRF but also to express CRF and its receptors [73]. Under stress conditions, CRF binds to its receptors located in hair follicle cells (DPCs and outer root sheath cells), which causes early catagen transition and the inhibition of hair shaft elongation. The molecular mechanisms of CRF action in DPCs involve increased intracellular concentration of reactive oxygen species (ROS) with subsequent apoptosis, cell cycle arrest at the G2/M phase, and downregulation of anagen-related cytokine levels. Consequently, increased CRF levels in hair follicle cells may result in hair loss [73,80]. Moreover, in the DPCs influenced by CRF, higher levels of other HPA hormones, such as cortisol, ACTH, and its precursor proopiomelanocortin (POMC), were observed. This effect may be connected with the upregulation of POMC gene expression and stimulation of POMC conversions by CRF [11]. This study also showed that CRF receptor antagonists blocked the HPA axis in the DPCs [73].

Dysfunctional cutaneous HPA axis might be one of the underlying causes of alopecia areata. Elevated CRH, ACTH, and α-melanocyte-stimulating hormone (α-MSH) levels were observed in HFs and the scalp epidermis of patients with AA [79,81]. Above all, CRH (along with substance P) may initiate AA. This involves two mechanisms: autoimmune and apoptotic. The autoimmune pathway dominates and is based on the collapse in the immune privilege of HFs. HFs begin to show a higher expression of major histocompatibility (MHC) class antigens and become a target for the immune system [76,79,82]. Some studies in vitro suggest a restoration of the immune privilege of HFs in the treatment of AA. They propose α-MSH, ACTH, and β-endorphin as immunomodulators acting through the suppression of the excessive expression of MHC class I [83]. Other studies suggest that the immunosuppressive function of ACTH and MSH may be restored thanks to UVA-1 treatment. The effect depends on the interruption of excessive CRH stimulation and the normalization of the HPA axis [84]. This field requires further studies [76].

Androgenetic alopecia is another hair loss type that might be connected with stress. Fischer et al. proved that CRH promoted a complex stress response in the HFs of patients with AGA. Researchers found increased expression of CRH receptors 1 and 2 (CRH-R1/2), ACTH, catagen-inducing transforming growth factor-β2 (TGF-β2), melanocortin receptor 2 (MC-R2), and other parameters associated with stress (substance P and p75 neurotrophin receptor). Moreover, in the HFs incubated with CRH, the expression of anagen-promoting insulin-like growth factor-1 (IGF-1) and matrix keratinocyte proliferation was significantly lower than in controls. Interestingly, the effects mentioned above were blocked by caffeine [76,85].

The role of cortisol in the pathogenesis of alopecia still remains unclear. It has been shown that high levels of cortisol can cause premature degradation and reduced synthesis of hyaluronan and proteoglycans located in the skin, which may negatively affect HFs [11,13,76,86]. Moreover, increased levels of cortisol were observed in patients with AGA compared to healthy individuals [13,87]. Thus, some studies put forward cortisol inhibitors as potential therapeutic options for patients with AGA [13,88,89]. Ketoconazole is an antifungal drug that blocks numerous enzymes, including 21-hydroxylase, a key enzyme in cortisol synthesis [90]. Additionally, it has antiandrogenic properties at high concentrations, thanks to the inhibition of enzymes involved in androgen synthesis and the competitive blockage of AR. Topical 2% ketoconazole provides a high local concentration in HF and allows for telogen-to-anagen transition [91,92]. Hence, topical ketoconazole is considered to be an alternative therapy for patients with AGA after first-line treatments of finasteride and minoxidil [89]. The target of this therapy is to stimulate proteoglycan synthesis and consequently counteract hair loss. Increased hair shaft diameter and hair regrowth were described in some studies; however, the evidence is limited, and further randomized controlled trials are needed in this field [13,89]. On the other hand, low cortisol levels can positively impact hair growth by delaying proteoglycan degeneration [11,76,86]. Furthermore, corticosteroids are used as a treatment in AA [93,94,95]. Topical corticosteroids and pulse dose corticosteroid therapy also show effectiveness in pediatric AA [96,97].

The assessment of cortisol levels may be a useful diagnostic tool in clinical practice [11,98]. Cortisol levels fluctuate during the day and can be measured in saliva samples [13,99].

Diurnal cortisol slope testing has not been particularly researched in relation to hair loss, although it may be used to evaluate the HPA axis’ balance and the impact of stress. Another promising method indicating chronic stress involves hair cortisol concentration measurements. Unfortunately, there are not enough data to currently confirm the utility of this method in patients with alopecia [13].

## 5. Thyroid Hormones

Thyroid hormones regulate numerous cellular activities, including the endogenous modulation of dermal proliferation and local inflammation [16,100]. They are also required for hair follicles’ physiological growth and maintenance [16]. During embryonic development, thyroid hormones control hair follicle differentiation and growth. THs impact both the anagen and telogen phases—they stimulate cell proliferation and metabolic activity within the HFs, which promotes the beginning and maintenance of the anagen phase; they also regulate the length of the telogen phase. These hormones also stimulate the production and distribution of melanin [100,101]. Hair loss is observed in patients with hypothyroidism or hyperthyroidism [100], but the hair condition is induced through different mechanisms. Hypothyroidism impedes epidermal and skin appendage cell division, decreasing anagen frequency [16,100]. On the other hand, hyperthyroidism results in oxidative damage. It increases free radical formation in mitochondria due to the stimulated production of reactive oxygen species. Still, the exact mechanism of its impact on hair loss remains not fully understood [100]. The thyroid profile is linked with the most common types of alopecia, including telogen effluvium (TE), alopecia areata (AA), androgenetic alopecia, and hair changes associated with thyroid malignancies, and those connections are elucidated below.

### 5.1. Autoimmune Thyroid Disorders

In patients with alopecia areata, there is a higher prevalence of autoimmune conditions, such as Hashimoto’s thyroiditis and Graves’s disease. Also, a higher incidence of anti-thyroid antibodies, including anti-thyroid peroxidase, antithyroglobulin, and TSH-receptor antibodies, is observed in this group of patients compared to the control. Approximately 5% of AA patients have subclinical hypothyroidism related to Hashimoto’s thyroiditis, and a significant proportion experience some form of thyroid dysfunction. Moreover, a female predominance is observed. There is also an immunological association between thyroid illnesses and AA—certain HLA types are connected to both AA and antibody-induced hypothyroidism. On the other hand, regulatory T cells are involved in both AA and thyroid condition progression. In line with this, some studies demonstrate a higher likelihood of AA in patients suffering from autoimmune hyperthyroidism or hypothyroidism as compared with healthy controls [100,102].

Autoimmune thyroid disorders may also be associated with MPA. For example, a study on 1232 men and women for 25 months identified 14.29% (*n* = 176) of them with MPA. The prevalence of thyroid dysfunctions in the MPA subgroup differed with age as follows: 6.3% in patients below 20 years old, 17% in persons aged between 21 and 40 years, and 50% in individuals older than 40 years [102].

### 5.2. Hypothyroidism and Hyperthyroidism

Unlike alopecia areata, telogen effluvium is linked to the thyroid hormones’ bloodstream levels and not to autoimmune conditions [102]. As T3 and T4 affect the follicles directly and make them sensitive during anagen, their low blood levels in hypothyroidism may lead to hair loss with prolonged shedding and also changes in hair texture [100]. One of the most recently published study results confirms the significant association between hypothyroidism and TE by pointing out the fact that hypothyroidism is a common underlying cause of TE. The prolonged telogen phase, associated with hypothyroidism, results in the typical hair loss seen in TE [103].

Moreover, a considerable prevalence of hypothyroidism is observed in patients with frontal fibrosing alopecia. According to data from a study conducted on patients with this type of hair loss, hypothyroidism was present in 22.9% of patients [104].

Some data indicate the connection between female pattern hair loss and hypothyroidism. In a prospective study conducted on FPHL patients aged 20 to 88 years, hypothyroidism was observed in 31.25% of individuals, but there was no correlation between the severity of hair loss and the presence of thyroid disease. More evidence is needed to define whether the increasing prevalence of thyroid disorders in this population is incidental or represents a part of a pathogenic mechanism [100,102].

According to the findings above, the thyroid profile should be checked in patients with hair loss to implement proper hormonal treatment in case of thyroid disorders and in order to improve alopecia treatment. The standard treatment for hypothyroidism is levothyroxine, administered orally, and after restoring normal thyroid function, it results in the effective relief of hypothyroidism symptoms in the majority of patients. In line with this, the hair loss caused by hypothyroidism usually improves within several months [105].

Hyperthyroidism may cause diffuse scalp hair thinning. Moreover, according to some authors, hair loss is observed in about 50% of hyperthyroid patients [100]. Hyperthyroidism treatment includes antithyroid drugs, radioiodine treatment, and surgery. Antithyroid drugs include methimazole and propylthiouracil, administered orally, usually for several months until the restoration of correct thyroid hormone levels. Upon achieving normal thyroid hormone levels, the underlying cause of hair loss disappears, and as a result, the condition of hair should improve. This may not happen if there is another coexisting pathology, including immune-mediated hair disorders that may be a complication of autoimmune hyperthyroidism [106]. On the other hand, side effects of antithyroid drugs include hair loss, which may further deteriorate the condition of hair. For this reason, therapy should always be individually targeted.

### 5.3. Hair Changes Associated with Thyroid Malignancies

There are a few potential ways in which hair may be affected when it comes to thyroid cancer. Individuals with thyroid malignancies may sometimes experience hair loss or thinning hair. This hair loss is not exclusive to the scalp and can also affect other body hair. Because of thyroid dysfunction in the course of thyroid cancer, changes in hair texture may be observed too [100,107]. Also, it is worth mentioning a systematic review of 123 individuals with histologically confirmed alopecia neoplastica (AN) that identified 7.3% (9/123) of them with a thyroid malignancy; 66.7% (6/9) were females [102]. Still, the amount of statistical evidence is low, and further investigation is needed for this junction to be better understood. Moreover, in rare cases, changes in hair color due to disrupted melanin production in patients with thyroid malignancies may be observed, which can involve a darkening or lightening of the hair [100,108].

## 6. Growth Hormone (GH)

Although growth hormone receptors have been found in the human hair follicle, the exact role of growth hormone in HF physiology remains unexplored. In general, in GH deficiency, dermatologic manifestations are observed, including alopecia, frontal hairline recession, and telogen effluvium. Moreover, it is associated with severe HF structural changes like *pili torti et canaliculi* and trichorrhexis nodosa [109,110,111]. In pathologies leading to GH deficiency, like Noonan syndrome, Turner syndrome, and Prader–Willi syndrome, alopecia, telogen effluvium, and frontline hair recession are observed [109,111]. Unexpectedly, ex vivo stimulation of GH receptors in human HFs results in hair growth inhibition in the female human scalp. Furthermore, primary decreases in GH and IGF-1 have been associated with hair loss and alopecia, while decreases in GH and IGF-1 due to GHRH deficiency have not [109,112,113].

Undoubtedly, a better understanding of the growth hormone effect on the HF, skin, and related signaling pathways (with potential sex differences taken into consideration) would be crucial to implementing new therapies for dermatopathology.

## 7. Conclusions

Hair loss, considering its significant prevalence, diverse causes, and both physical and psychological impact, should be understood as a serious health problem that requires a proper diagnostic process. It can also be a sign of underlying internal disease, including disorders of the endocrine system, in particular, the thyroid, ovaries, testes, adrenal glands, and pituitary gland, as discussed in this review. We believe that the link between this dermatological manifestation and endocrine disorders cannot be ignored, and hair loss investigation may lead to the correct diagnosis of the underlying endocrine disease, thereby allowing for proper treatment to be provided and allowing for improvement of both hair status and the internal condition of the patient.

## Figures and Tables

**Table 1 biomedicines-12-00513-t001:** Hormonal changes in hair loss.

Hormone	Hormone Level	Possible Endocrine Disorder	Type of Alopecia
**Androgens**			
T DHT A4 DHEA DHEAS	increased	androgen excess polycystic ovary syndrome * congenital adrenal hyperplasia * androgen-producing tumors	androgenetic alopecia
SHBG	decreased		
FAI **	increased		
**Estrogens**			
E2	decreased	menopause	female pattern hair loss frontal fibrosing alopecia
**Stress hormones**			
CRH	increased	chronic stress	stress-induced alopecia alopecia areata
ACTH	increased/decreased ***	Cushing disease	androgenetic alopecia
Cortisol	increased		
**Thyroid hormones**			
T3 T4	decreased/increased	hypothyroidism/hyperthyroidism	telogen effluvium alopecia areata androgenetic alopecia
**Others**			
GH	decreased	GH deficiency Turner syndrome Noonan syndrome Prader–Willi syndrome	androgenetic alopecia telogen effluvium

Abbreviations: T—testosterone, DHT—dihydrotestosterone, A4—4-androstenedione, DHEA—dehydroepiandrosterone, DHEAS—dehydroepiandrosterone-sulfate, SHBG—sex hormone-binding globulin, FAI—free androgen index, E2—estradiol, CRH—corticotropin-releasing hormone, ACTH—adrenocorticotropic hormone, T3—triiodothyronine, T4—thyroxine, GH—growth hormone. * In the differentiation between polycystic ovary syndrome and congenital adrenal hyperplasia, 17-hydroxyprogesterone (17-OHP) is useful (increased in congenital adrenal hyperplasia vs. normal or slightly increased in PCOS). ** Free androgen index (FAI) = total testosterone level/sex hormone-binding globulin level. *** Increased in pituitary adenoma; decreased in adrenal tumor.

**Table 2 biomedicines-12-00513-t002:** Hormonal treatments in hair loss.

Medication	Mechanism of Action	Route of Administration	Dose	Indication
Finasteride	5α-reductase type II inhibitor	oral	1 mg daily	Androgenetic alopecia
		topical	0.25% solution (1 mL twice daily)	
Dutasteride	5α-reductase type I and II inhibitor	oral	0.5 mg daily	
Spironolactone	AR * antagonist, 17α-hydroxylase inhibitor	oral	25–200 mg daily	
		topical	1% gel or 5% solution twice daily	
Bicalutamide	AR antagonist	oral	10–50 mg daily	
Cyproterone acetate	5α-reductase inhibitor, AR antagonist, gonadotrophin secretion inhibitor	oral	50 mg daily	
Clascoterone	AR antagonist	topical	1% cream	
Pyrilutamide	AR antagonist	topical	0.5% solution (1 mL once/twice daily)	
Pumpkin seed oil	Herbal 5α-reductase inhibitor	oral	400 mg daily	
Estradiol	ER agonist, elevation of sex hormone-binding globulin levels	oral	1–2 mg daily	menopause, premature ovarian failure
Levothyroxine	Synthetic version of human thyroxine	oral	25–200 µg daily	hypothyroidism

* AR—androgen receptor.

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
