# Peer review of "The Hormonal Background of Hair Loss in Non-Scarring Alopecias"

_biomedicines, 2024, doi:10.3390/biomedicines12030513_

Round 1

Reviewer 1 Report

Comments and Suggestions for Authors

I have read the review "Hormonal background of hair loss". I have the following comments and suggestions:

- The title properly reflects the subject of the paper.

- The authors have systematically presented the information and have made their case correctly.

- The writing is clear and concise, and the figure and tables are appropriate and informative and enhance the manuscript.

- However, the authors only discuss hormonal influence for non-scarring types of alopecia. Is there any known influence of hormones on different types of scarring alopecias (e.g. Lichen planopilaris, central centrifugal cicatricial alopecia, frontal fibrosing alopecia, folliculitis decalvans, DLE, etc.)? Authors can add that info to the manuscript. If they don't want to do that, they can consider changing the name of the manuscript to "Hormonal background of hair loss in non-scarring alopecias."

Thanks!

Comments on the Quality of English Language

Minor editing is required.

Author Response

We would like to thank the Reviewer for the careful and thorough reading of this paper. We sincerely appreciate the constructive and valuable comments. We revised the manuscript and uploaded the revised version. We would like to address the Reviewer's comments below.

I have read the review "Hormonal background of hair loss". I have the following comments and suggestions:

- The title properly reflects the subject of the paper.

- The authors have systematically presented the information and have made their case correctly.

- The writing is clear and concise, and the figure and tables are appropriate and informative and enhance the manuscript.

- However, the authors only discuss hormonal influence for non-scarring types of alopecia. Is there any known influence of hormones on different types of scarring alopecias (e.g. Lichen planopilaris, central centrifugal cicatricial alopecia, frontal fibrosing alopecia, folliculitis decalvans, DLE, etc.)? Authors can add that info to the manuscript. If they don't want to do that, they can consider changing the name of the manuscript to "Hormonal background of hair loss in non-scarring alopecias."

Thank you for this comment. We considered that writing about the hormonal reasons for scarring alopecias would be beyond the scope of this article. Therefore, we changed the title according to your suggestion.

We also thoroughly reviewed the English language in the manuscript, and ultimately, it underwent proofreading by a native English speaker.

Reviewer 2 Report

Comments and Suggestions for Authors

This is a comprehensive review of hair loss from endocrinopathies. The authors do not only focus on AGA, but all types of hair loss, both in men and women, which makes the paper significantly more useful to clinicians than papers focusing only on one type of hair loss.

I have some comments:

1. Please define serious (abstract) and significant (introduction) psychological distress. Is higher risk of depression, anxiety, low-self-esteem etc a serious psychological distress? Are there existing definitions on seriousness/significance of distress? Perhaps only use 'serious distress' in abstract and introduction.

2. I suggest you provide information on what grade 3 on H-N scale means. Otherwise, the information is not useful for the clinician.

3. Why do you not present treatment options for hyperthyroidism, which also causes hair loss and which is mentioned as a cause earlier? Or is the ROS-induced hair loss irreversible? Or is restoring normal levels of thyroxin not efficacious?  If those are the reasons for not giving treatment options for those cases, this also should be mentioned.

4. What did the combined therapy group receive besides spironolactone?

Minor:
- Line 407: The abbreviation AN is not explained previously
- Some places you use corticotropin, other places corticotrophin when describing CRH. Please use the same throughout.
- Typo: pili torti er canaliculi - change to 'et'

Author Response

We would like to thank the Reviewer for the careful and thorough reading of this paper. We sincerely appreciate the constructive and valuable comments. We revised the manuscript and uploaded the revised version. We would like to address Reviewer’s comments below.

This is a comprehensive review of hair loss from endocrinopathies. The authors do not only focus on AGA, but all types of hair loss, both in men and women, which makes the paper significantly more useful to clinicians than papers focusing only on one type of hair loss.

I have some comments:

  1. Please define serious (abstract) and significant (introduction) psychological distress. Is higher risk of depression, anxiety, low-self-esteem etc a serious psychological distress? Are there existing definitions on seriousness/significance of distress? Perhaps only use 'serious distress' in abstract and introduction.

Thank you very much for this comment. We agree that these two words may sound confusing. Here, we meant high levels of psychological distress, but unfortunately, there is no existing definition of the seriousness/significance of distress, and in numerous clinical studies, different terms were used, i.e serious, severe, significant. Therefore, as you suggested we only used 'serious distress' in abstract and introduction.

  1. I suggest you provide information on what grade 3 on H-N scale means. Otherwise, the information is not useful for the clinician.

We described grade 3 on H-N scale in the revised manuscript on page 3, based on the original decription.

‘At stage III, the first signs of clinically significant balding appear. According to the original description by Norwood et al., most type III scalps have deep frontotemporal recessions which are usually symmetrical and are either bare or very sparsely covered by hair. These recessions extend further posteriorly than a point which lies approximately 2 cm anterior to a coronal line drawn between the external auditory meatus [35,36].’

  1. Why do you not present treatment options for hyperthyroidism, which also causes hair loss and which is mentioned as a cause earlier? Or is the ROS-induced hair loss irreversible? Or is restoring normal levels of thyroxin not efficacious?  If those are the reasons for not giving treatment options for those cases, this also should be mentioned.

We added information about treatment options for hyperthyroidism on page 10, as restoring normal levels of thyroid hormones usually improves hair condition:

‘Hyperthyroidism may cause diffuse scalp hair thinning. Moreover, according to some authors, hair loss is observed in about 50% of hyperthyroid patients [100]. Hyperthyroidism treatment includes antithyroid drugs, radioiodine treatment, and surgery. Antithyroid drugs include methimazole and propylthiouracil, administered orally, usually for several months until the restoration of correct thyroid hormone levels. Upon achieving normal thyroid hormone levels, the underlying cause of hair loss disappears, and as a result, the condition of hair should improve. This may not happen if there is another coexisting pathology, including immune-mediated hair disorders that may be a complication of autoimmune hyperthyroidism [106]. On the other hand, side effects of antithyroid drugs include hair loss, which may further deteriorate the condition of hair. For this reason, the therapy should always be individually targeted.’

  1. What did the combined therapy group receive besides spironolactone?

The combined therapy group received oral spironolactone and minoxidil. We added that information in the revised manuscript (page 5).

Minor:
- Line 407: The abbreviation AN is not explained previously
- Some places you use corticotropin, other places corticotrophin when describing CRH. Please use the same throughout.
- Typo: pili torti er canaliculi - change to 'et'

Thank you for these comments, these mistakes were corrected.